# Characterization of Cortical and Subcortical Structural Brain Asymmetry in Adults with and without Dyslexia

**DOI:** 10.3390/brainsci13121622

**Published:** 2023-11-23

**Authors:** Jacqueline Cummine, Tiffany Ngo, Kelly Nisbet

**Affiliations:** 1Department of Communication Sciences and Disorders, Faculty of Rehabilitation Medicine, University of Alberta, Edmonton, AB T6G2G4, Canada; tngo4@ualberta.ca (T.N.); kanisbet@ualberta.ca (K.N.); 2Neuroscience and Mental Health Institute, Faculty of Medicine, University of Alberta, Edmonton, AB T6G2G4, Canada

**Keywords:** reading, subcortical, caudate, putamen, thalamus, asymmetry, lateralization

## Abstract

Multiple cortical (planum temporale, supramarginal gyrus, fusiform gyrus) and subcortical (caudate, putamen, and thalamus) regions have shown different functional lateralization patterns for skilled vs. dyslexic readers. The extent to which skilled and dyslexic adult readers show differential structural lateralization remains to be seen. Method: Participants included 72 adults (N = 41 skilled; N = 31 dyslexic) who underwent a high-resolution MRI brain scan. The grey matter volume of the cortical and subcortical structures was extracted. Results: While there were clear behavioral differences between the groups, there were no differences in any of the isolated structures (i.e., either total size or asymmetry index) and limited evidence for any brain–behavior relationships. We did find a significant cortical–cortical relationship (*p* = 0.006) and a subcortical–subcortical relationship (*p* = 0.008), but not cross-over relationships. Overall, this work provides unique information on neural structures as they relate to reading in skilled and dyslexic readers.

## 1. Introduction

To date, there has been considerable work that explores the lateralization of multiple cortical regions (e.g., superior temporal sulcus; supramarginal gyrus; fusiform gyrus) involved in skilled and impaired reading populations. More recently, there is a growing consensus that subcortical brain structures also play a significant role in various aspects of language [1] and motor [2] processes. Indeed, there is much evidence that activity among the subcortical structures of the brain is highly influenced by sensory, motor, and cognitive demands [1,3,4]. To date, the structural properties of these cortical and subcortical regions, as they relate to reading ability/disability, are less well understood. In a recent review by Janacsek et al. [1], a theoretical framework, known as the MaMa dynamic network model of brain and cognition, was proposed to further investigate the interplay between subcortical and cortical brain regions. What is particularly noteworthy about this review is that the caudate, putamen, and thalamus were all implicated in reading (both skilled and dyslexic; [1] see Figures 2 and 3). As such, we focus here on these three structures that have been consistently implicated in reading [1,4,5] and speech production [2,6] and examine the extent to which the volume and/or asymmetry of these structures is differentially implicated in skilled vs. dyslexic readers.

### 1.1. Lateralization of Subcortical Structures Involved in Reading

Caudate. The caudate is a structure embedded in the striatum [3], a major input nucleus to the cortical–subcortical network where complex information is initially processed. In the speech production literature, the caudate is implicated in speech planning [2] and has been noted for playing a role in rhythm. In line with this, Fondas et al. [7] reported that children who stutter (CWS; a neurodevelopmental disorder characterized by atypical speech planning and execution processes, see also [8]) had reduced volume in the right caudate compared to children who did not stutter. In addition, they also found an increased leftward asymmetry in the caudate of CWS compared to controls. Notably, reading involves the processes of speech planning, particularly for novel stimuli such as nonwords; therefore, these caudate asymmetries may also exist between differential reading groups. Indeed, with respect to reading processes, caudate activity has been particularly implicated in reading impairment related to sound sequencing and word decoding (i.e., nonword reading performance). For example, Cheema et al. [9] found that during nonword reading, for adults with reading impairments, the activity in the caudate was significantly correlated with the activity in the thalamus (a subcortical structure implicated in motor plans, to be discussed further below). That is, dyslexic readers demonstrated a significant coordination or planning loop [2] between these subcortical structures during difficult reading tasks (see also Krafnick et al. [10], who showed increased activity in the right caudate following a reading intervention for school-aged children). In the current study, we aim to explore whether these previous findings extend to (1) adults with and without reading impairments and (2) the structural properties of the caudate.

Putamen. Together, the caudate and putamen comprise the dorsal striatum [3]. The putamen is a structure with many projections to the motor/premotor cortex, and thus it is not surprising that the putamen has been implicated in both word reading [9] and speech production [6]. In the speech production literature, the putamen is ascribed to motor planning and initiation [2] and forms part of the motor loop with the thalamus. In line with this, the functional connectivity between the putamen and other brain regions has been shown to be aberrant in adults who stutter (AWS) [8]. From a structural perspective, Beal et al. [11] reported that children who stutter (CWS) had reduced volume in the left putamen compared to children who did not stutter, whereas Lu et al. [12] showed that adults who stutter had greater volume in the left putamen compared to adults who do not stutter. As fluent reading involves the processes of fluent speech production, particularly for familiar stimuli (i.e., real words), these volumetric putamen patterns may also be present/distinct between individuals with and without reading impairments. In line with this notion, Cheema et al. [9] found that during real word reading, for adults with and without reading impairments, the activity in the putamen was significantly correlated with the activity in the thalamus. That is, readers demonstrated a significant coordination or motor loop [2] between these subcortical structures during automatized and familiar reading tasks (see also Muyler et al. [5], who showed increased activity in the putamen and thalamus following a reading intervention for school-aged children). In a similar vein to the literature on the caudate, we have yet to determine if these previous findings extend to the structural properties of the putamen in individuals with and without reading impairments.

Thalamus. Finally, the thalamus is a crucial part of the cortical–subcortical network, as it has reciprocal projections to both the cortical and subcortical regions. In general, the thalamus is known as a major relay station among the subcortical structures, passing along (or inhibiting) both sensory and motor information to other cortical and subcortical structures [2,3,12]. In the speech domain, adults who stutter show decreased activity in the thalamus compared to their nonstuttering counterparts [12]. In the reading domain, thalamic connectivity has been shown to be related to nonword behavioral performance in children [13] and letter–word identification in young adults [14]. As mentioned above, Cheema et al. [9] reported a caudate–thalamus connection for nonword reading and a putamen–thalamus connection for real word reading, suggesting that the thalamus plays an important role in the common motor representations found in both of these tasks. Finally, in a meta-analysis of brain activity following intervention for reading impairments, the left thalamus was reported as a structure that showed increased activity post-treatment [15]. As such, for all structures, it is important to consider right vs. left differences, in addition to asymmetry of the structure, given the impact of reading impairment on language regions over the course of development and into adulthood (see Perdue et al. [16] for a review that outlines the neural correlates of reading impairment (and intervention), with a particular focus on describing normalization of function (i.e., left hemisphere) vs. compensatory function (i.e., right hemisphere)). 

### 1.2. Lateralization of Cortical Structures Involved in Reading

Contrary to the work on the subcortical structures as they relate to reading, there is considerably more research investigating the differential lateralization of the planum temporale, supramarginal gyrus, and fusiform gyrus between skilled and impaired readers. For example, a meta-analysis found significant reductions in the brain volume of individuals with reading impairment compared to typically developing individuals in the left superior temporal sulcus and left superior temporal gyrus that were related to functional differences as well [17]. A more recent meta-analysis by Eckert et al. [18] similarly highlighted the important differences in volume of the left superior temporal sulcus as a possible locus of differences between individuals with and without reading impairments. Regions in the occipito-temporal and supramarginal gyri have also been highly noted in dyslexia research (i.e., decreased grey matter for individuals with dyslexia) [19,20,21,22,23,24,25]; however, there remains considerable inconsistency in the reported effects and, to date, there is limited research that includes adults with reading impairments (see [19] for a recent review). Overall, these papers demonstrate the importance and value of examining brain structure differences underlying dyslexia and signify a gap in the literature with respect to adult participants.

### 1.3. Summary

In summary, there is a large body of work pointing to the important role that subcortical structures play in reading and speech production. To date, the majority of the work has focused on the functional properties of these structures (i.e., brain activity in response to stimuli and/or intervention), with much less information about the structural properties of these structures. Here we address this gap in the literature by examining the volumetric properties of the caudate, putamen, and thalamus in individuals with and without reading impairment, and the extent to which these subcortical structures are related to cortical reading areas and/or reading behavior. 

Research questions: Are there differences between skilled and dyslexic adult readers in the structural properties of the cortical and subcortical structures?Are there relationships between reading behavior and subcortical structure asymmetry?Are there relationships between the cortical and subcortical regions?

## 2. Methods

The data were collected as part of two separate studies (see [26,27,28] for additional study information). Relevant to the current study, a total of 72 participants were included (N = 41 skilled readers; Females = 20; Mean Age = 21; SD = 2.4; Mean Education Level = 15.3 years; N = 31 dyslexic readers; Females = 23; Mean Age = 23; SD = 4.2; Mean Education Level = 16.4 years). Inclusion criteria for the skilled group consisted of English as the native or primary language, normal or corrected-to-normal vision, no contraindications to the MRI, and age-appropriate scores on reading and IQ measures. Inclusion criteria for the dyslexic group consisted of English as the native or primary language, normal or corrected-to-normal vision, no contraindications to the MRI, and age-appropriate scores on the nonverbal IQ test (Note: For sample [26], skilled reader (M = 83%) and dyslexic reader (M = 81%) participants did not differ with respect to nonverbal IQ (*p* = 0.548). The full dataset for nonverbal IQ was not available for the study [28]). In addition, participants in the dyslexic group had to score at least 1 SD below the skilled group on a standardized reading task (see [28] for additional details on the battery of behavioral measures; see [29] for a similar classification approach) and score at or above 0.45 on the Adult Reading History Questionnaire [30]. Exclusion criteria for both groups included a history of any hearing or vision impairment, as well as stroke and/or any neurological disorders like ADHD. All participants were paid an honorarium of CAD 30 cash for their participation. The data reported here were collected as part of a larger study that was approved by the University of Alberta Research Ethics Board (Pro00135696), and all participants provided their informed consent.

### 2.1. Behavioral Data Collection

All participants were administered the following tasks. Participants completed the Sight Word Efficiency (SWE) subtest and the Pseudo-Word Decoding Efficiency (PDE) subtest of the Test of Word Reading Efficiency—1st Edition (TOWRE) [31] (Note: behavioral data were missing for 7 skilled readers. A mean substitution was used in these cases; TOWRE SWE = 106; TOWRE PDE = 100). Measures extracted included standardized scores (i.e., accuracy) for reading real words and nonwords, respectively.

### 2.2. MRI Acquisition

For N = 38 participants (22 skilled and 16 dyslexic readers; 18 Females; Mean age = 21.2), images were acquired on a 1.5 T Siemens Sonata scanner (Siemens, Medical Systems, Erlangen, Germany) and were positioned along the anterior–posterior commissure line. Anatomical scans included a high-resolution axial T1 MPRAGE sequence with the following parameters: TR = 2000 ms; TE = 4.38 ms; number of slices = 112; base resolution = 256 × 256; voxel size = 1 × 1 × 1 mm; and scan time = 4:48 min. 

For N = 34 participants (19 skilled and 15 dyslexic readers; 25 Females; Mean age = 22.6), images were acquired on a 3.0 T Siemens MAGNETOM Prisma scanner and were positioned along the anterior–posterior commissure line. Anatomical scans included a high-resolution axial T1 MPRAGE sequence with the following parameters: repetition time (TR) = 1700 ms; echo time (TE) = 2.21 ms; number of slices = 176; base resolution 256 × 256 × 176; with a voxel size of 1 × 1 × 1 mm; and scan time of 4.50 min. 

### 2.3. Segmentation Measurements

Fully automated segmentation of the brain was performed using the vol2brain program (https://www.volbrain.net/; Accessed on 1 August 2022; [32]), which implements a multi-atlas patch-based label fusion segmentation. Single anonymized compressed MRI T1w Nifti files are uploaded through a web interface (Figure 1A). Preprocessing consists of de-noising, rough inhomogeneity correction, affine registration to the MNI space, fine inhomogeneity correction, and intensity normalization (see [32] for complete details). Importantly, these preprocessing steps are optimized to minimize between-scanner variability [33]. To estimate the structural properties of the participants’ brains, a nonlocal intra-cranial cavity extraction was conducted to create a brain mask. Tissue classification into white matter, grey matter, and cerebral spinal fluid was conducted on this mask. The pipeline segmented the brain into 135 structures to determine the cortical volumetric measurements (Figure 1B). For the purposes of this study, we were interested in: (1) the total cortical volume of the fusiform gyrus, planum temporale, supramarginal gyrus, and (2) the total subcortical volume of the caudate, putamen, and thalamus. For each region, the volume was normalized against the total brain volume for each participant (i.e., volume in cm^3^/total brain volume × 100). In addition, the asymmetry index was extracted, which is calculated as the difference between right and left volumes divided by their mean (in percent) (https://www.volbrain.net/pages/tutorials; accessed on 1 August 2022). Negative values correspond to a greater leftward asymmetry and positive values correspond to a greater rightward asymmetry. 

### 2.4. Analysis

RQ1: Are there differences between skilled and dyslexic readers in the structural properties of the cortical or subcortical structures? A series of independent t-tests were performed to test for differences between skilled and dyslexic readers on each of the structural dependent variables (total volume and asymmetry; a total of 12 tests). In the interest of minimizing Type 2 error, we report Bonferroni corrected and uncorrected *p*-values to assess significance.

RQ2: Are there relationships between behavior (TOWRE real words and TOWRE nonwords) and structure asymmetry? Nonparametric Spearman’s correlations were run for each behavior (6 structures × 2 behaviors = 12 correlations). In the interest of minimizing Type 2 error, we report Bonferroni corrected and uncorrected *p*-values to assess significance.

RQ3: Are there structural connections among the cortical and subcortical regions? Nonparametric Spearman’s correlations were run among the structures on the asymmetry index and collapsed across group classification. In the presence of significant relationships, follow-up Fisher tests were run to determine whether there was a significant difference in r-value magnitude between the groups.

## 3. Results

Behavioral results. The skilled readers performed significantly better on the TOWRE PDE task (Mean 100; SD 10.9) compared to the dyslexic readers (Mean 82; SD 10.3), t(70) = 7.33, *p* < 0.001 (Cohen’s d = 1.75). The skilled readers performed significantly better on the TOWRE SWE task (Mean 106; SD 9.7) compared to the dyslexic readers (Mean 83; SD 9.5), t(70) = 9.97 *p* < 0.001 (Cohen’s d = 2.37).

RQ1: Are there differences between skilled and dyslexic readers in the structural properties of the cortical or subcortical structures?

A series of independent samples t-tests show that there are no differences between skilled and dyslexic readers in the total volume or asymmetry values for any of the structures (Table 1).

RQ2: Are there relationships between behavior (TOWRE real words and TOWRE nonwords) and structure asymmetry?

There was a single significant correlation between thalamus asymmetry and TOWRE PDE, whereby stronger leftward asymmetry values were associated with higher TOWRE PDE scores (*p* = 0.035; Figure 2). This relationship was not significant for either group alone. There were no other significant relationships between behavior and structure asymmetry (Table 2).

RQ3: Are there structural connections among the cortical and subcortical regions?

There was a significant positive correlation between asymmetry of the subcortical thalamus and caudate, whereby stronger leftward asymmetry of one structure was associated with stronger leftward asymmetry of the other structure (*p* = 0.008; Figure 3). There was a significant negative correlation between asymmetry of the cortical planum temporale and supramarginal gyrus, whereby stronger leftward asymmetry of one structure was associated with stronger rightward asymmetry of the other structure (*p* = 0.006; Figure 4) (Table 3).

## 4. Discussion

In this work, we sought to explore the structural asymmetry properties of the cortical and subcortical regions as they relate to skilled and dyslexic reading performance. Overall, we found no significant differences between skilled and dyslexic readers in the size or lateralization of the structures themselves. We also found little evidence for relationships between cortical/subcortical asymmetry and behavior, apart from a single relationship between thalamus asymmetry and nonword reading performance. Finally, we did find support for: (1) a cortical–cortical relationship between the planum temporale and supramarginal gyrus, and (2) a subcortical–subcortical relationship between the thalamus and caudate. We discuss the implications for these findings in the context of neural correlates of reading and reading impairment.

### 4.1. Structural Asymmetry and Reading Performance: Brain–Behavior Relationships

Previous work has provided functional evidence for the role of the caudate in nonword/decoding tasks [9,34], and for the role of the thalamus in the fluency and initiation of motor programs [2]. Here, we found little evidence that these relationships extended into the structural domain. Specifically, we found a single relationship between thalamus asymmetry and nonword reading performance; however, this relationship was quite weak and would not survive a correction for Type 1 error. There were no other relationships found between reading performance and any of the cortical or subcortical structures. However, we note that three findings associated with the caudate approached significance, namely the difference between reading groups on caudate asymmetry (*p* = 0.067), and the relationship between caudate asymmetry and nonword reading (*p* = 0.072), and real word reading (*p* = 0.063). The convergence of these trends on the caudate does warrant some caution in oversimplifying the lack of findings in the current work. Nonetheless, the overall ‘null’ effects provide important information about the neural signatures associated with reading and reading impairment. That is, the behavioral manifestations associated with reading performance appear to be more strongly associated with neural activity and not the neural structure. This is further evidenced by the lack of differences in structure asymmetry between skilled and impaired readers. 

Our findings are also in contrast to findings reported in children [18] and other neurodevelopmental disorders [7]. In these works, volumetric differences were found between the groups of interest (i.e., dyslexia and children who stutter). The current findings suggest that the structural differences found in younger groups of individuals are not static but in fact continue to be modified throughout development and ultimately result in similar neural–structural profiles between adults with and without neurodevelopmental disorders. What is particularly notable is that the functional activity between these groups of individuals continues to show divergent patterns in adulthood [1,9]. The MaMa model described in Janacsek et al. [1] provides several pathways to account for these functional connectivity differences, which is a necessary avenue of inquiry as we further investigate the neural profiles of reading ability and disability in adults. Furthermore, the contribution of the subcortical structures and their connection with cortical structures needs much more dedicated work as it remains to be seen how/if the structural properties of these regions also provide unique information about reading processes.

### 4.2. Asymmetry of Structure

We extend previous work on the lateralization of function [16] via the examination of the lateralization of the structure. Interestingly, we found that the asymmetry of the planum temporale was positively correlated with the asymmetry of the supramarginal gyrus. That is, as one region became more left-lateralized, so too did the other region. We also found that the asymmetry of the caudate and the thalamus was negatively correlated, such that greater leftward asymmetry of one structure was related to more rightward asymmetry of the other structure. These two findings, namely cortical–cortical and subcortical–subcortical, have not yet been reported in the literature for either skilled or dyslexic readers and provide unique information on the potential co-development of these structural regions. Future work that examines functional connectivity between these regions is also needed to provide further characterization of how the brain networks contribute to skilled and impaired reading processes.

### 4.3. Limitations

Importantly, our work focused on the extraction of the volume of the entire structure of interest. However, it is well documented that each of the structures explored here can be further segmented into smaller units of interest. Each of the smaller units contain distinct connections with subcortical and cortical regions, unique sensitivity to neurotransmitters (i.e., dopamine; [7]), and unique interneurons/neurons. [3] The extent to which these finer-grained structures differentially relate to reading ability/disability should be further explored (i.e., see Ali et al. [35] for a demonstration of the role of the head of the caudate in word interference).

Further, given that the study was a secondary analysis of previously collected data, there were several factors outside of our control that may have contributed to variability in the data and resulted in nonsignificant findings. Most relevant are the scanner strength (i.e., 1.5 T vs. 3.0 T) and asymmetry of gender representation (i.e., proportionally more females in the 3.0 T study compared to the 1.5 T study). While we provided Appendix A that speak to the (lack of) differences in brain structure between the two scanner strengths, ultimately an interaction between gender and scanner strength cannot be fully ruled out. Ultimately, future studies that aim to balance demographic, biological, and experimental factors will be important to support/refute the null effects reported here. 

## 5. Conclusions

There is a large body of work pointing to the important role that cortical and subcortical regions play in reading and speech production. Here, we provide evidence that the structure of these regions is not different between skilled and impaired readers, nor do these structures relate to behavioral performance in any significant manner. However, there were significant cortical–cortical and subcortical–subcortical connections that provide evidence for structural connectivity in regions associated with reading. Overall, cortical and subcortical structures are necessary to include in working neuroanatomical models of reading ability/disability, and the extent to which the structural integrity of these regions subserve core computations for lower- and higher-order reading processes [1] remains to be seen.

## Figures and Tables

**Figure 1 brainsci-13-01622-f001:**
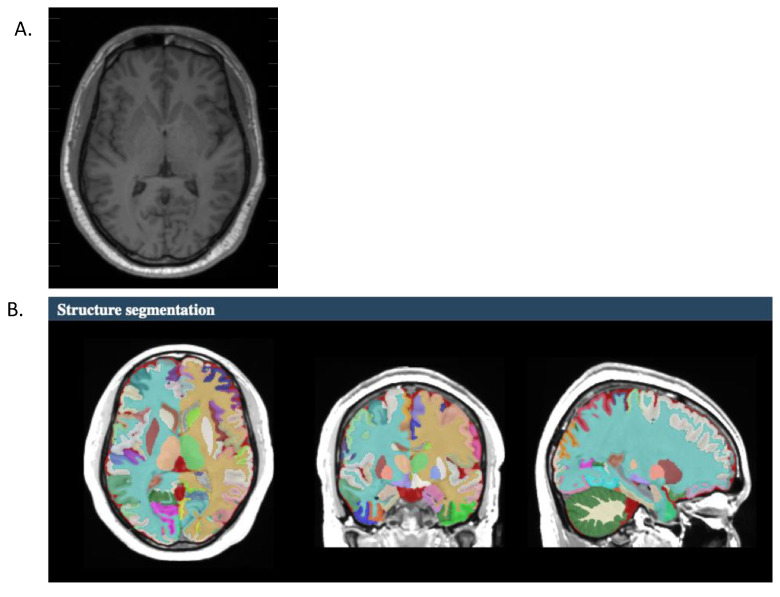
Representative brain image for one participant. (**A**). High-resolution anatomical image submitted to volbrain2.0 for processing. (**B**). Structural segmentation of the high-resolution anatomical image.

**Figure 2 brainsci-13-01622-f002:**
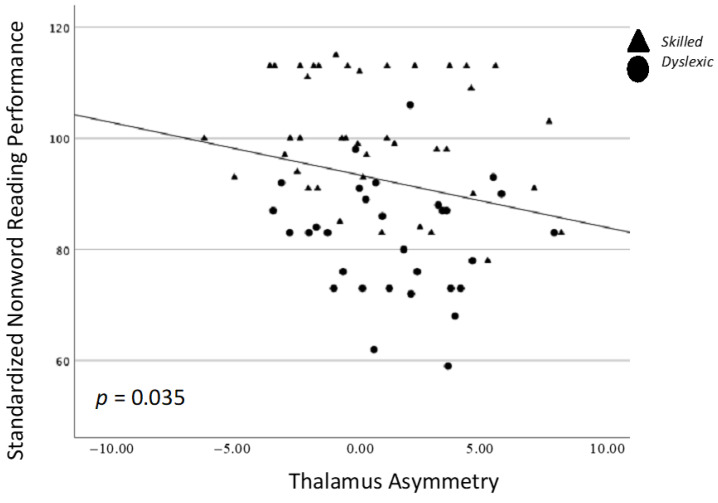
Spearman’s correlation between nonword reading performance (TOWRE PDE) and the asymmetry of the thalamus.

**Figure 3 brainsci-13-01622-f003:**
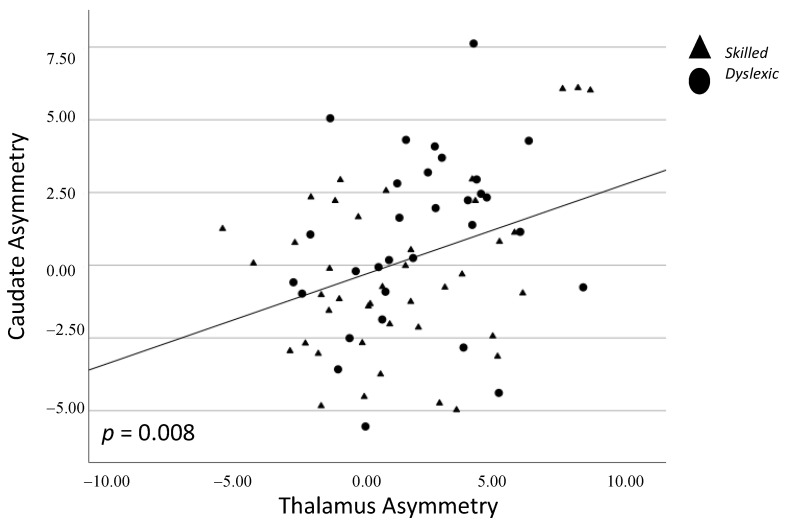
Spearman correlation between caudate and thalamus asymmetry. Triangles = skilled; Circles = dyslexic.

**Figure 4 brainsci-13-01622-f004:**
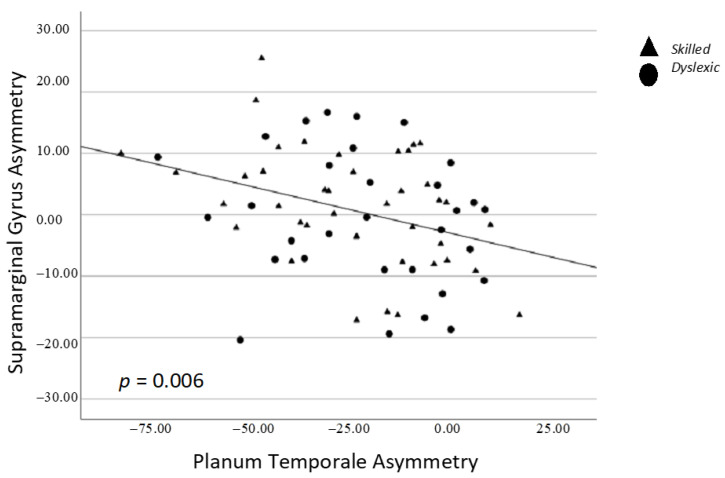
Spearman correlation between supramarginal gyrus and planum temporale asymmetry. Triangles = skilled; Circles = dyslexic.

**Table 1 brainsci-13-01622-t001:** The mean volumetric values for each group. ^+^ Positive = rightward asymmetry. Negative = leftward asymmetry.

	Structure	Skilled	Dyslexic	*p*-Value
Subcortical	Caudate Total (cm^3^) ^+^ Asymmetry (%)	7.87 (0.94) −0.36 (2.85)	7.63 (1.10) 0.91 (2.96)	0.640 0.067
	Putamen Total (cm^3^) ^+^ Asymmetry (%)	9.07 (0.93) −0.42 (2.41)	8.69 (1.08) −0.18 (2.47)	0.537 0.681
	Thalamus Total (cm^3^) ^+^ Asymmetry (%)	12.89 (1.13) 0.79 (3.40)	12.44 (1.31) 1.58 (2.74)	0.645 0.288
Cortical	Fusiform Gyrus Total (cm^3^) ^+^ Asymmetry (%)	16.9 (2.50) −3.21 (12.1)	16.9 (3.08) −1.67 (10.6)	0.797 0.575
	Supramarginal Gyrus Total (cm^3^) ^+^ Asymmetry (%)	18.2 (1.99) 1.42 (9.5)	18.1 (3.24) −0.61 (10.9)	0.800 0.400
	Planum Temporale Total (cm^3^) ^+^ Asymmetry (%)	4.23 (0.81) −16.8 (22.7)	4.15 (1.08) −13.3 (22.6)	0.712 0.518

**Table 2 brainsci-13-01622-t002:** Spearman’s coefficients (*p*-value) that represent the relationship between behavior and asymmetry. * *p* < 0.05 uncorrected.

	Caudate	Putamen	Thalamus	Fusiform Gyrus	Planum Temporale	Supramarginal Gyrus
TOWRE SWE	−0.220 (0.063)	0.019 (0.875)	−0.129 (0.280)	−0.132 (0.269)	−0.058 (0.627)	0.140 (0.241)
TOWRE PDE	−0.213 (0.072)	−0.018 (0.880)	−0.249 (0.035) *	−0.095 (0.430)	0.014 (0.905)	0.025 (0.837)

**Table 3 brainsci-13-01622-t003:** Spearman’s coefficients (*p*-value) that represent the relationship between cortical and subcortical structures. * *p* < 0.05 uncorrected.

	Caudate	Putamen	Thalamus	Fusiform Gyrus	Planum Temporale
Caudate	-				
Putamen	−0.131 (0.272)	-			
Thalamus	0.311 * (0.008)	−0.046 (0.700)	-		
Fusiform Gyrus	0.183 (0.123)	0.096 (0.422)	−0.068 (0.571)	-	
Planum Temporale	−0.016 (0.895)	−0.038 (0.750)	−0.055 (0.645)	−0.229 (0.053)	-
Supramarginal Gyrus	0.003 (0.982)	0.025 (0.832)	0.037 (0.759)	0.108 (0.367)	−0.318 * (0.006)

## Data Availability

The data presented in this study are available on request from the corresponding author. The data are not publicly available due to ethical restrictions.

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
