# Peer review of "Characterization of Cortical and Subcortical Structural Brain Asymmetry in Adults with and without Dyslexia"

_brainsci, 2023, doi:10.3390/brainsci13121622_

Round 1

Reviewer 1 Report

Comments and Suggestions for Authors

In the introduction, it is imperative that the research team cites contemporary literature and that all the reported data relates to the existing knowledge on the issue. In 1.1, the reference of those related to dyslexia and the profile of dyslexics is well described for the brain structures associated with the dyslexic brain.

All bibliography references are explicit and naturally guide the reader into the topic. I propose that 1.3 can be integrated with 1.4.

The research questions, hypotheses, and methodology used are transparent. My only objection, which is not understandable why it was chosen, is the different use of MRI acquisition, especially since the two machines that were used do not have the same protocol. It is not distinguishable by which criteria one or the other patient was included in the corresponding MRI machine and why there are two different MRI procedures. Based on this fact, I would expect the results to describe and compare the two groups separately since we have subgroups of dyslexic and typical readers. However, this does not happen; the results are presented as a single group for each sample.

The measurements, the analyses made, and the statistics chosen serve the purpose and answer the research questions in a structured way.

The Spearman correlations are evident. Perhaps with a slightly larger sample, it would have had more substantial statistical significance in the measurements that showed a positive correlation.

The discussion is step-by-step outlined, and the correlation of the results is documented and linked to the existing literature and the variation that exists in other studies, which offers something to the scientific community and should be studied in future research (I think the writing team rightly emphasizes this in its conclusions).

The team notes the limitations and future implications of the study. The entire structure follows the standards of the journal as well as the citation of the bibliography. It is an excellent study with a careful and substantiated presentation of knowledge.

Author Response

In the introduction, it is imperative that the research team cites contemporary literature and that all the reported data relates to the existing knowledge on the issue. In 1.1, the reference of those related to dyslexia and the profile of dyslexics is well described for the brain structures associated with the dyslexic brain.

Response:

Thank you. We have attempted to capture current and relevant literature to the questions at hand.

All bibliography references are explicit and naturally guide the reader into the topic. I propose that 1.3 can be integrated with 1.4.

Response:

We agree with the reviewer. This is a formatting process undertaken at the editing level of the journal. We would support this recommendation.

The research questions, hypotheses, and methodology used are transparent. My only objection, which is not understandable why it was chosen, is the different use of MRI acquisition, especially since the two machines that were used do not have the same protocol. It is not distinguishable by which criteria one or the other patient was included in the corresponding MRI machine and why there are two different MRI procedures. Based on this fact, I would expect the results to describe and compare the two groups separately since we have subgroups of dyslexic and typical readers. However, this does not happen; the results are presented as a single group for each sample.

Response:

Thank you for your thoughtful comment. We have updated the manuscript and now include supplementary material to this point. In brief, the data was a secondary analysis of previously collected data, which resulted in the different magnet strengths. We have done some additional exploration into the extent to which the magnet strength produced differing results and the impact is minimal. Where the difference was present (i.e., thalamus), follow up t-tests did not show any difference between skilled and impaired readers. However, we have also updated the limitations section to caution the reader with respect to these factors.

The measurements, the analyses made, and the statistics chosen serve the purpose and answer the research questions in a structured way.

The Spearman correlations are evident. Perhaps with a slightly larger sample, it would have had more substantial statistical significance in the measurements that showed a positive correlation.

The discussion is step-by-step outlined, and the correlation of the results is documented and linked to the existing literature and the variation that exists in other studies, which offers something to the scientific community and should be studied in future research (I think the writing team rightly emphasizes this in its conclusions).

The team notes the limitations and future implications of the study. The entire structure follows the standards of the journal as well as the citation of the bibliography. It is an excellent study with a careful and substantiated presentation of knowledge.

Response:

Thank you very much for your thoughtful consideration of our work.

Reviewer 2 Report

Comments and Suggestions for Authors

The subject of the paper is interesting and there really are many scientific gaps on this topic. The introduction included the necessary items for the subject, but highlighting the aspects of neuroimaging as a whole (activation, connectivity) in addition to morphometry, the focus of the article. Highlighting three specific questions for analysis. However, some methodological precautions were not very well taken to ensure the results presented were safe. One of them is that the images were collected on different equipment and with different magnetic fields as well, which can interfere with the quality of the image and its results. With this, I would like to know if the authors verified whether this factor influenced the result, showing the analysis independently, even if it is included in supplementary material, I think this is very relevant. Another important factor is that when we analyze morphometry of isolated regions (cortical or subcortical) we have to normalize these images to the total volume of the brain since we are making a comparison between groups and not within groups. And I did not find this methodological care in the text. If this was done, how was the participants' intracerebral volume normalized for comparison between segments?

In Figure 2, it would be interesting to insert the definition of the groups in the image, or include in the caption, which group is representing the triangle and the circle, to make it easier for the reader to analyze the result, as was done in Figures 3 and 4.

There was a lack of demographic description of the subjects included in the study, as we do not know the average, minimum and maximum age of the adults nor their gender and education, or the distribution of the non-verbal intelligence assessment applied, relevant factors for the dysfunction under analysis. And whether these factors were balanced between samples from the same magnetic resonance equipment or between them.

I think the authors could also explore these demographic issues a little more, since the difference between genders is so relevant to brain morphometry, it could be that they are failing to detect subtle differences with the mixture of so much noise (magnetic field, gender, education, among other factors, lack of normalization of brain volume, .....). From these other analyzes the discussion can be improved and much more enriching, contributing even more to understanding the scientific gaps in this topic.

Comments on the Quality of English Language

Moderate, I suggest a careful review

Author Response

Reviewer 2:
The subject of the paper is interesting and there really are many scientific gaps on this topic. The introduction included the necessary items for the subject, but highlighting the aspects of neuroimaging as a whole (activation, connectivity) in addition to morphometry, the focus of the article. Highlighting three specific questions for analysis. However, some methodological precautions were not very well taken to ensure the results presented were safe. One of them is that the images were collected on different equipment and with different magnetic fields as well, which can interfere with the quality of the image and its results. With this, I would like to know if the authors verified whether this factor influenced the result, showing the analysis independently, even if it is included in supplementary material, I think this is very relevant. 

Response:

We now include a supplementary Table that includes the independent samples t-tests for between scanner comparisons. see attachment.

​​

Another important factor is that when we analyze morphometry of isolated regions (cortical or subcortical) we have to normalize these images to the total volume of the brain since we are making a comparison between groups and not within groups. And I did not find this methodological care in the text. If this was done, how was the participants' intracerebral volume normalized for comparison between segments?

Response:

Thank you for this recommendation. We have gone back to our data, normalized each participant and brain region to account for their total brain volume, and re-run our statistics. The p-values in Table 1 now reflect the independent samples t-tests on the normalized brain volumes. 

We updated our analysis section to include the following information: “For each region, the volume was normalized against the total brain volume for each participant (i.e., volume in cm3/total brain volume * 100).” 

In Figure 2, it would be interesting to insert the definition of the groups in the image, or include in the caption, which group is representing the triangle and the circle, to make it easier for the reader to analyze the result, as was done in Figures 3 and 4.

Response:

Thank you for pointing out this error. We have updated the figure accordingly.

There was a lack of demographic description of the subjects included in the study, as we do not know the average, minimum and maximum age of the adults nor their gender and education, or the distribution of the non-verbal intelligence assessment applied, relevant factors for the dysfunction under analysis. And whether these factors were balanced between samples from the same magnetic resonance equipment or between them.

I think the authors could also explore these demographic issues a little more, since the difference between genders is so relevant to brain morphometry, it could be that they are failing to detect subtle differences with the mixture of so much noise (magnetic field, gender, education, among other factors, lack of normalization of brain volume, .....). From these other analyzes the discussion can be improved and much more enriching, contributing even more to understanding the scientific gaps in this topic.

Response:

We now report the breakdown of gender and age by scanner. In addition, we have added the supplementary table which reports no difference between the scanners with respect to age, reading scores, or volumes (apart from the thalamus). We have also updated the limitations section to speak directly to these potential influences.

“For N = 38 participants (22 skilled and 16 dyslexic readers; 18 Females; Mean age = 21.2), For N = 34 participants (19 skilled and 15 dyslexic readers; 25 Females; Mean age = 22.6).”

Further, given that the study was a secondary analysis of previously collected data, there were several factors outside of our control that may have contributed variability to the data, and resulted in non-significant findings. Most relevant are the scanner strength (i.e., 1.5T vs. 3.0T) and asymmetry of gender representation (i.e., proportionally more females in the 3.0T study compared to the 1.5T study). While we provided supplementary data that speaks to the (lack of) differences in brain structure between the two scanner strengths, ultimately an interaction between gender and scanner strength cannot be fully ruled out. Ultimately, future studies that aim to balance demographic, biological and experimental factors will be important to support/refute the null effects reported here. “